# Impact of Poloxamer on Crystal Nucleation and Growth of Amorphous Clotrimazole

**DOI:** 10.3390/pharmaceutics15082164

**Published:** 2023-08-21

**Authors:** Jie Zhang, Ziqing Yang, Liquan Luo, Kang Li, Taotao Zi, Junjie Ren, Lei Pan, Ziyue Wang, Zihao Wang, Minzhuo Liu, Zhihong Zeng

**Affiliations:** College of Biological and Chemical Engineering, Changsha University, Changsha 410022, China; zhangjie448215@163.com (J.Z.);

**Keywords:** poloxamer, nucleation, crystallization, polymorphs, clotrimazole

## Abstract

Surfactants have been widely used as effective additives to increase the solubility and dissolution rates of amorphous solid dispersions (ASDs). However, they may also generate adverse effects on the physical stability of ASDs. In this study, we systematically investigated the impacts of poloxamer, a frequently used surfactant, on the crystallization of amorphous clotrimazole (CMZ). The added poloxamer significantly decreased the glass transition temperature (*Tg*) of CMZ and accelerated the growth of Form 1 and Form 2 crystals. It was found that the poloxamer had an accelerating effect on Form 1 and Form 2 but showed a larger accelerating effect on Form 1, which resulted from a combined effect of increased mobility and local phase separation at the crystal–liquid interface. Additionally, the added poloxamer exhibited different effects on nucleation of the CMZ polymorphs, which was more complicated than crystal growth. The nucleation rate of Form 1 was significantly increased by the added poloxamer, and the effect increased with increasing P407 content. However, for Form 2, nucleation was slightly decreased or unchanged. The nucleation of Form 2 may have been influenced by the Form 1 crystallization, and Form 2 converted to Form 1 during nucleation. This study increases our understanding of poloxamer and its impacts on the melt crystallization of drugs.

## 1. Introduction

In recent years, amorphous solid dispersions (ASDs) have been the preferred approach for delivering water-insoluble drugs [1,2,3,4]. In ASDs, polymeric materials serve as carriers in which the drugs are dispersed at the molecular level [1,2,3,4]. Amorphous pharmaceutics exhibit higher solubility and faster dissolution than crystalline materials due to the higher Gibbs free energy compared with that of the crystalline counterpart [1,2,3,4]. The Food and Drug Administration (FDA) has approved more than 20 commercial ASD products [1,5].

However, the higher Gibbs free energy induces crystallization of amorphous drugs during the manufacturing processes, storage, and dissolution. To prepare solid dispersions, the selection of a polymer and an excipient is of great importance because it influences the manufacturing process, physical stability, dissolution, and bioavailability of the ASDs [6,7,8]. Surfactants have been reported to increase the dissolution rates and solubility of poorly soluble drugs [9,10,11]. Meanwhile, the surfactants in ASDs act as plasticizers and can be used in the melting process to decrease processing temperature or reduce the melt viscosity of formulations, thus helping to facilitate thermal processing via melt extrusion [12]. It was reported that the highest reduction of 50℃ was observed for Kollidon^®^ SR with plasticizers such as Lutrol^®^ F68, Cremophor^®^ RH40, and PEG 1500 used during the hot-melt extrusion process [13]. However, the surfactant leads to decreased *Tg* and increased molecular mobility due to the plasticizing effect, thus influencing the physical stability of the ASD [14]. Only a few previous studies have discussed the effects of surfactants on crystallization [15,16]; thus, the mechanism is not well understood.

Poloxamer is a triblock copolymer (poly(ethyleneoxide)-poly(propylene oxide)-poly(ethylene oxide)) containing semicrystalline PEO segments and amorphous PPO segments. It is available under a variety of trade names such as Pluronic^®^ from BASF; thus, it was found that poloxamer P407 (P407) and Pluronic^®^ F-127 are essentially the same polymer [12,17,18]. It can be used as a polymeric carrier in ASDs and has a significant effect on amorphous drugs [19,20,21,22,23]. Qian and coworkers evaluated the impacts of a poloxamer on the crystallization and microstructure of an ASD [22,23]. Because the poloxamer was a semicrystalline polymer, the amorphous drug crystallized within the semicrystalline matrix. The authors found that drug distribution was affected by the properties of the drug and the drug–polymer intermolecular interactions [22,23]. Yao et al. found that poloxamer could accelerate crystal nucleation and growth of amorphous nifedipine by a similar factor under the condition of no poloxamer crystallization [21]. Poloxamer is used more often as a surfactant in ASDs. For instance, Vasconcelos et al. evaluated the impact of poloxamer 407 on the bioavailability of the resveratrol Soluplus^®^ (1:2) ASD. Their in vivo results showed that the *AUC_o-t_* and *C_max_* of the ASD containing 15% P407 were 2.5-fold higher than those of the ASD without the poloxamer [9].

In this study, we used clotrimazole (CMZ) as a model drug and investigated the impact of P407 on the crystallization of CMZ. CMZ is an antimycotic imidazole agent and is commonly used for studying crystallization [24,25]. Two polymorphs (Form 1 and Form 2) of CMZ have been reported [24,25]. The metastable Form 2 can only be obtained from melt crystallization and may transform to Form 1 during the melt crystallization [24,25]. In the present study, the plasticizing effect of P407 was investigated through the change in the glass transition temperature (*Tg*). Meanwhile, the nucleation and crystal growth rates of CMZ polymorphs were also measured as a function of temperature or the content of P407. The phase separation and crystal transformation were observed and had a great influence on the crystallization of the CMZ polymorphs.

## 2. Materials and Methods

### 2.1. Materials

Clotrimazole was purchased from Bide Pharmatech Co., Ltd. (Shanghai, China) (purity > 99.0%, Form 1). Poloxamer 407 was obtained from Basf Chemicals Co., Ltd. (Shanghai, China) (Figure 1).

### 2.2. Preparation of CMZ/P407 Physical Mixtures and ASDs

CMZ/P407 physical mixtures containing different contents of P407 (2.5%, 5%, 10%, and 20% *w*/*w*) were prepared with a mortar and pestle. To prepare ASD, 2–5 mg CMZ/P407 physical mixtures were melted at 155 °C between two round glass coverslips, and then quenched to room temperature.

### 2.3. Polarized Light Microscopy (PLM)

Crystalline morphologies were observed by PLM (Guangzhou Micro-Shot Technology, Guangzhou, China), and a hot stage (Shangguang (Suzhou) Instrument, Suzhou, China) and oven were used for temperature control.

### 2.4. Powder X-ray Diffraction (PXRD)

Crystallinity was determined with a Bruker D8 Advance with Cu Kα radiation (λ = 1.542 Å). The samples were placed on a zero-background silicon sample holder and scanned from 10 to 40° at a speed of 10°/min.

### 2.5. Differential Scanning Calorimetry (DSC)

DSC was performed with a Q2000 system (TA Instruments, New Castle, DE, USA) and carried out under 50 mL/min nitrogen purge. A 3–5 mg sample was weighed and loaded in an aluminum pan. The sample was first heated at 10 °C/min to 160 °C for 1 min, and then cooled at 20 °C/min to 0 or −20 °C; then, it was reheated to a temperature above the melting temperature at 10 °C/min.

### 2.6. Crystal Growth Rates of CMZ

Each 2–5 mg sample was melted at 155 °C between two round glass coverslips, and then quenched to room temperature. The amorphous samples were held for several minutes or hours at room temperature for nucleation of Form 1 and 2, and then transferred to high temperature for growth rate determination. The growth rates were measured as previously described [24]. All determinations were performed in triplicate.

### 2.7. Nucleation Rates of CMZ

The nucleation rates were determined by the two-stage method. Nuclei were formed at the desired temperature, and then the crystals grew to visible sizes at a high temperature. In the present study, the sample weighed 2–5 mg and was melted above the melting point, quenched to a desired temperature, and held for different times to enable nucleation, as described in our previous studies [24]. The nucleation rates were expressed as the number of nuclei per unit volume as a function of time. The volume could be determined by film thickness and field of view. The film thickness was calculated by the sample mass, CMZ and P407 density, and the film surface area. All determinations were performed in triplicate.

## 3. Results and Discussion

### 3.1. State of Mixing between CMZ and P407

Figure 2a shows DSC traces for CMZ (Form 1) samples containing different P407 contents ranging from 0 to 20% *w*/*w*. The Tm of P407 is near 50 °C, and the heat fusion increases with increasing P407 content. The Tm of the pure drug is near 140 °C and decreases with increasing P407 content. A phase diagram for CMZ and P407 was constructed based on this work and reported data [26]. Pure Form 2 is hard to obtain from a solution or melt, and Form 2 can be transformed to Form 1 in the solid state; thus, the phase diagram of Form 2 and P407 cannot be constructed using this method. The freshly prepared CMZ/P407 ASDs are optically clear and exhibit no birefringence under PLM, suggesting that CMZ is miscible with P407.

Meanwhile, the CMZ/P407 ASDs prepared by melt quenching exhibited a single *Tg*. Figure 3 shows the *Tg* values for CMZ samples containing 0–20% *w*/*w* P407. The *Tg* values of the ASDs decreased with increasing P407 content. Previous studies reported that the *Tg* of P407 is −69 and −66 °C [21,27]; thus, the molecular mobility of CMZ was increased after mixing with the low-*Tg* polymer [23].

### 3.2. Crystallizations of CMZ Doped with P407

The crystal nucleation and growth of pure CMZ has been evaluated previously [24]. Figure 4 compares the morphologies of CMZ polymorphs grown from pure CMZ and from CMZ doped with different P407 contents at 70 °C. For CMZ containing different contents of P407, the Form 1 crystals grew as compact spherulites and were not significantly affected by low P407 contents. For the ASDs with higher P407 contents (5% and 10% *w*/*w*), a transparent layer formed at the crystal–liquid interface at 70 °C (Figure 4 and Appendix A). The transparent layer indicated local phase separation, which was described in our previous study [28,29]. The transparent layer was a polymer-enriched phase, which crystallized at room temperature and melted at 60 °C due to the high content of P407, as shown in Appendix A. For the samples containing 20% *w*/*w* P407, the Form 2 material grew as a single crystal.

The CMZ polymorph grown from pure CMZ or CMZ containing different P407 contents was studied by PXRD (Figure 5). The samples were nucleated at 50 °C and then grown at 80 °C. The characteristic peaks for both CMZ polymorphs were found in the CMZ doped with 2.5–10% *w*/*w* P407, and no new peaks were present in the PXRD results, which was consistent with the PLM results. For CMZ doped with 20% *w*/*w* P407, only characteristic peaks for Form 1 were observed, which may be attributed to the low content of Form 2 in the sample.

Figure 6 shows the morphologies of CMZ crystals grown from pure CMZ or CMZ doped with different contents of P407 after nucleation at 50 °C for 14,100 s, followed by growth at 80 °C. In comparing the nucleation processes of the CMZ polymorphs, Figure 6 shows that the number of Form 2 polymorphs was much larger than that of Form 1 polymorphs for the pure drug; however, as the P407 content was increased, the number of Form 1 polymorphs increased and that of Form 2 decreased, indicating that the relative nucleation rates of Forms 1 and 2 changed as the P407 content increased. This will be quantitatively investigated in the following sections.

Figure 7a shows bulk crystal nucleation rates (*J*) for CMZ polymorphs with and without 10% *w*/*w* P407. P407 significantly increased the nucleation rates of Form 1 and showed minimal effects on Form 2, or even decreased the nucleation rates at 50 °C. For neat amorphous CMZ, nucleation of the Form 2 crystals was significantly faster than that of Form 1. However, for samples doped with 10% *w*/*w* P407, nucleation of Form 1 crystals was faster than that of Form 2. To our knowledge, no study has focused on the effects of additives on the nucleation of polymorphs; thus, it is not well understood. The selectivity for CMZ polymorph nucleation will be discussed in the next sections.

Figure 7b shows the bulk crystal growth rates (*u*) of the CMZ polymorphs with and without 10% *w*/*w* P407. The crystal growth of Form 2 was significantly faster than that of Form 1, and the difference increased with cooling. With the addition of P407, the relative growth rates for CMZ polymorph nucleation did not change. Figure 7b shows that P407 accelerated the crystal growth of Forms 1 and 2, but this effect was smaller for Form 2 than Form 1. It was reported that the accelerating effects depended on the polymer content at the crystal–liquid interface [28,29]. In a previous study, the PEO exhibited a selective effect on the crystallization of indomethacin polymorphs [28]. The accelerating effects and PEO concentration at the crystal growth front decreased in the same order: γ form > α form > δ form. It was proposed that the polymer content was correlated with the solubility of the IMC polymorphs in the polymer [28]. As in the previous study, the polymer had larger effects on the stable form during crystallization. Based on the solubility of the drug in the polymer and the polymer content at the growth front, we concluded that more polymers were present in the growth front of the stable form than in the metastable form; thus, the polymer has a greater effect on the stable form.

Figure 8a,b show the growth kinetics of CMZ crystal Forms 1 and 2 containing different amounts of P407 at 50 and 80 °C. The crystal growth rates for two polymorphs showed similar trends with increasing P407 content. The crystal growth rates reached plateaus when the P407 loading reached 5% *w*/*w* at 50 °C. In a comparison of the crystal growth kinetics (Figure 8) with the *Tg* values, the *Tg* values decreased continuously with increasing surfactant content, whereas the growth rates plateaued.

The growth rate plateau phenomenon was discussed in our previous study and is affected by local phase separation at the growth front [28,29]. In this study, interestingly, the growth rate plateau decreased at 80 °C with increasing polymer contents above 10% *w*/*w*. Crystal growth is influenced by both thermodynamic and kinetic factors [30]. At higher temperatures, molecular diffusivity is higher, and crystallization is faster. At lower temperatures, increased supercooling (*T–Tm*) exerts a greater thermodynamic driving force for crystallization. Therefore, kinetics favor crystallization at higher temperatures, while thermodynamics favor crystallization at lower temperatures. Thus, if both factors are considered, the crystal growth rates will reach at a maximum somewhere between *Tg* and *Tm* due to the complex interplay of thermodynamic and kinetic factors. In this study, the thermodynamic factor had a greater impact. The growth rates were determined by thermodynamics at temperatures closer to *Tm*, which was suppressed by the P407 content (shown in Figure 2 and Appendix A).

Figure 9 shows the crystal nucleation kinetics at 50 °C for CMZ Form 1 and Form 2 containing different amounts of P407. With increasing surfactant content, the crystal nucleation rate for Form 1 increased. Yu and coworkers investigated the effects of surfactants on the nucleation of nifedipine [21]. They found that four surfactants showed similar effects on nucleation due to their roles as mobility enhancers [21]. In this study, the enhancement of Form 1 was attributed to the mobility increase caused by P407. However, for Form 2, the results showed that the addition of P407 decreased the nucleation rates, which cannot be explained by the mobility. This will be discussed in the following sections.

### 3.3. Relationship between Mobility and Crystallization Kinetics

The effects of low-*Tg* polymer on the crystallization of organic compounds have been studied previously [31,32]. Sato et al. investigated the effects of a biocompatible polymer with a low *Tg* on the growth rates of a variety of organic compounds [31]. They evaluated the effects of different polymer contents on the crystallization rates by normalizing them with the *Tg* (*T*–*Tg*; *T* is the crystallization temperature) [31]. With polymer concentrations of 5% or 10%, they found that the growth rates for compounds with various polymer contents vs. (*T*–*Tg*) were similar to those for the pure drug, which was attributed to mass transport effects [31]. In the present study, as shown in Figure 10, we also normalized the nucleation and growth rates for CMZ with 10% *w*/*w* P407 added as a function of *T–Tg*. For the nucleation rates, the diffusion zone of Form 1 was nearly superimposable, while that of Form 2 was not. The growth rates for both Form 1 and Form 2 were nearly superimposable. This suggested that polymer enrichment at the growth front had a relatively minor effect on crystallization, and the plasticizing effects on the molecular dynamics were more prominent for the nucleation and growth of Form 1. Additionally, it seemed that other factors could impact the nucleation of Form 2, not just mass transport effects.

### 3.4. Impact of P407 Enrichment on Crystallization

As shown in Figure 11 and Appendix A, a transparent layer was formed at the Form 1 crystal front when it grew to a certain size. However, the transparent layer of Form 1 affected the growth of Form 2 when it approached Form 2. The transparent layer was reported to have a polymer content higher than that of the bulk material [28]. The higher polymer content decreased the melting point of the crystals, as shown in Figure 2. It has been reported that the polymer content at the crystal growth front influenced the accelerating effects: a higher content induced stronger acceleration. This result is consistent with the accelerating effects shown in Figure 6 and Figure 7. In this study, the polymer content at the Form 1 crystal front was higher than that for Form 2; thus, the transparent layer could induce melting of the Form 2 crystals.

Recently, several studies showed that nucleation rates can be predicted by the crystal growth kinetics because the two processes have similar activation barriers [21,33]. The authors found that the nucleation rate could be predicted with the equation *J*/*J*_0_
*= u*/*u*_0_, where *u* is the crystal growth rate, and *J*_0_ and *u*_0_ are the nucleation and growth rates of the pure drug at a given temperature *T*.

In this study, we also used this model to predict the nucleation rates of CMZ doped with P407. The predicted *J* values were plotted against the observed *J* values and compared with those from other systems; the points from previous studies were located around the diagonal line (Figure 12). This plot suggested that the additives influenced the two processes to similar extents. The results for Form 1 supported the view that nucleation and growth are both mobility-limited and that the surfactant served as a plasticizer and caused similar accelerations of the two processes [21]. Form 2 showed a significant deviation between our predicted and observed *J* values. The observed nucleation rate of Form 2 is lower than that predicted from the growth rate, indicating that factors other than mobility influenced nucleation. Appendix A shows that the transformation rates of Form 2 into Form 1 were significantly enhanced by P407. We speculate that the Form 2 crystals would convert to Form 1 during nucleation rate determination. This result suggests that predictions of nucleation rates from growth rates should be made carefully, especially for multicomponent systems with multiple polymorphs.

## 4. Conclusions

We investigated the impact of poloxamer on the crystallization of amorphous clotrimazole. The poloxamer acted as a plasticizer and significantly decreased the glass transition temperature of the clotrimazole. It was found that the poloxamer increases the growth rate of two polymorphs of clotrimazole and has a greater effect on Form 1. Additionally, the poloxamer showed a selective effect on the nucleation rate of the clotrimazole polymorphs. The effects of the poloxamer on crystal growth resulted from a combined effect of mobility and local phase separation. The nucleation rate for Form 1 was increased significantly by the poloxamer, while the nucleation of Form 2 was slightly decreased or unchanged. The observed nucleation rate of Form 2 was lower than that predicted from the growth rate, which may be attributed to the transformation of Form 2 during nucleation. The transformation rates of Form 2 into Form 1 were significantly enhanced by P407; thus, P407 may induce Form 2 crystals to convert to Form 1 in the early stages of nucleation. This suggests that predictions of nucleation rates of the metastable form from growth rates should be made carefully for multicomponent systems with multiple polymorphs. This study provides insight into the impacts of surfactants on the nucleation and growth kinetics of drugs during melt crystallization.

## Figures and Tables

**Figure 1 pharmaceutics-15-02164-f001:**
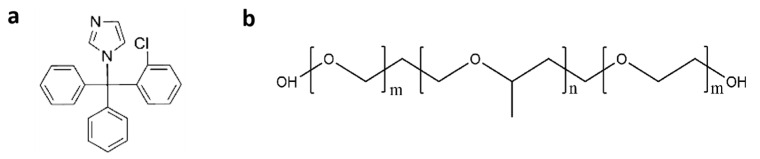
Structures of CMZ (**a**) and P407 (**b**).

**Figure 2 pharmaceutics-15-02164-f002:**
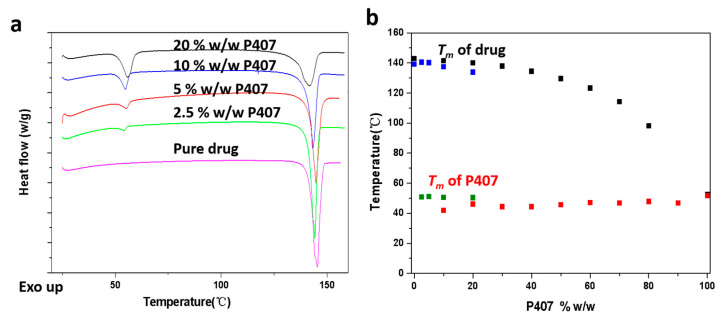
DSC traces of CMZ-P407 mixtures (**a**). Onset of CMZ Form 1 and P407 melting as a function of the drug weight fraction (black squares and red squares indicate the melting points of the drug and P407. Reproduced with permission from Ref. [26], Pharmacotherapy Group, 2014; the data presented with blue squares (drug) and green squares (P407) were obtained in this study) (**b**).

**Figure 3 pharmaceutics-15-02164-f003:**
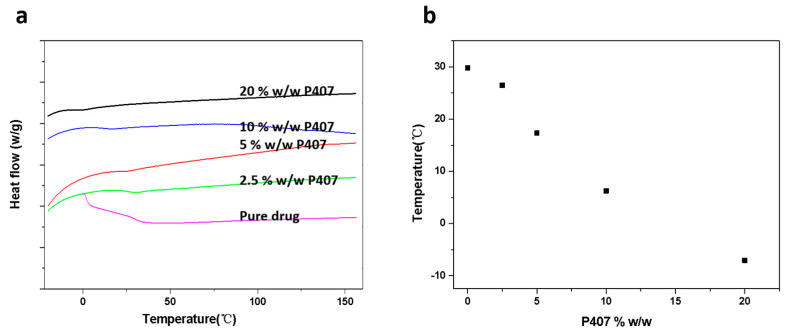
DSC traces of CMZ-P407 mixtures during heating (**a**); *Tg* values of amorphous CMZ samples containing 0–20% *w*/*w* P407 (**b**).

**Figure 4 pharmaceutics-15-02164-f004:**
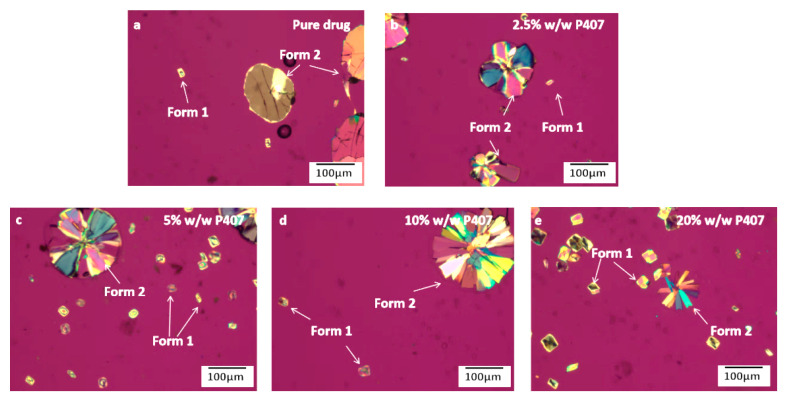
Crystal morphologies of pure CMZ or CMZ grown in the presence of different P407 contents at 70 °C, pure drug (**a**); 2.5% *w*/*w* P407 (**b**); 5% *w*/*w* P407 (**c**); 10% *w*/*w* P407 (**d**); 20% *w*/*w* P407 (**e**).

**Figure 5 pharmaceutics-15-02164-f005:**
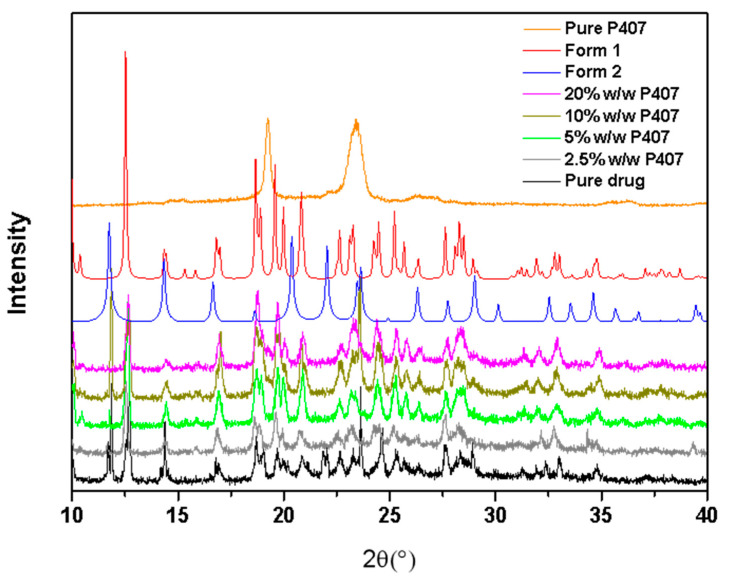
PXRD patterns of CMZ containing different P407 contents, nucleated at 50 °C, and grown at 80 °C. The PXRD of Forms 1 and 2 was simulated from CCDC files 130537 and 2222298, respectively.

**Figure 6 pharmaceutics-15-02164-f006:**
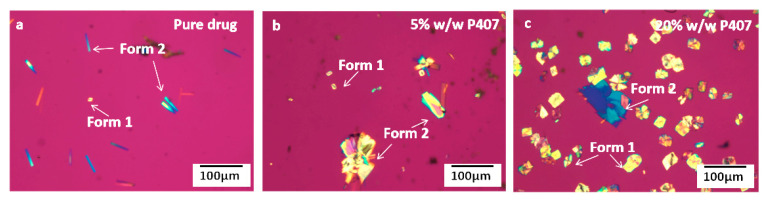
Form 1 and Form 2 obtained in CMZ doped with different P407 contents: pure drug (**a**), 5% *w*/*w* P407 (**b**), and 20% *w*/*w* P407 (**c**) after nucleation at 50 °C for 14,100 s, followed by growth at 80 °C.

**Figure 7 pharmaceutics-15-02164-f007:**
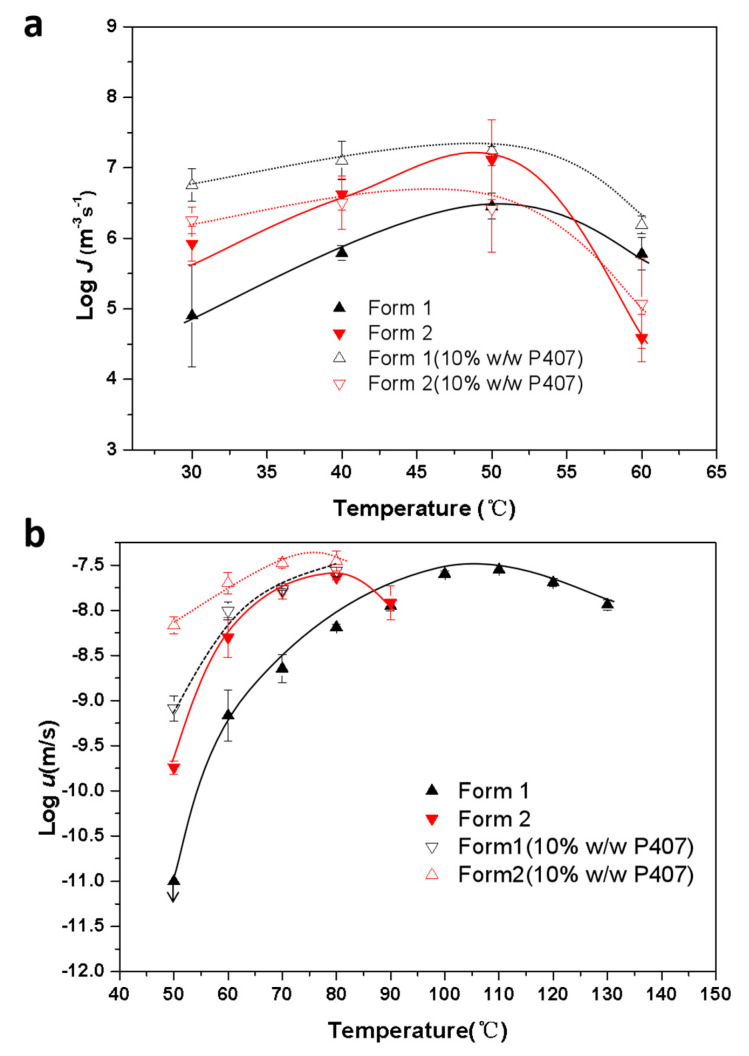
Crystal nucleation kinetics (**a**) and crystal growth kinetics (**b**) for amorphous CMZ in the absence and presence of 10% *w*/*w* P407 (the curves are only a guide to the eye).

**Figure 8 pharmaceutics-15-02164-f008:**
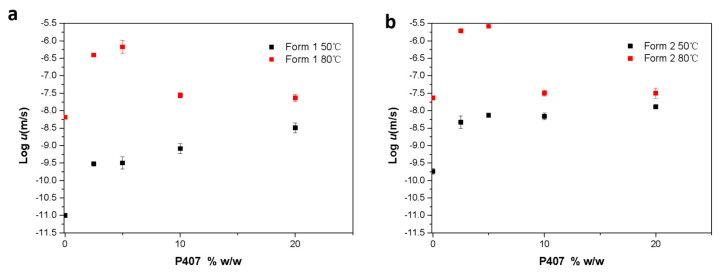
Crystal growth kinetics for CMZ Form 1 (**a**) and Form 2 (**b**) as a function of the P407 concentration.

**Figure 9 pharmaceutics-15-02164-f009:**
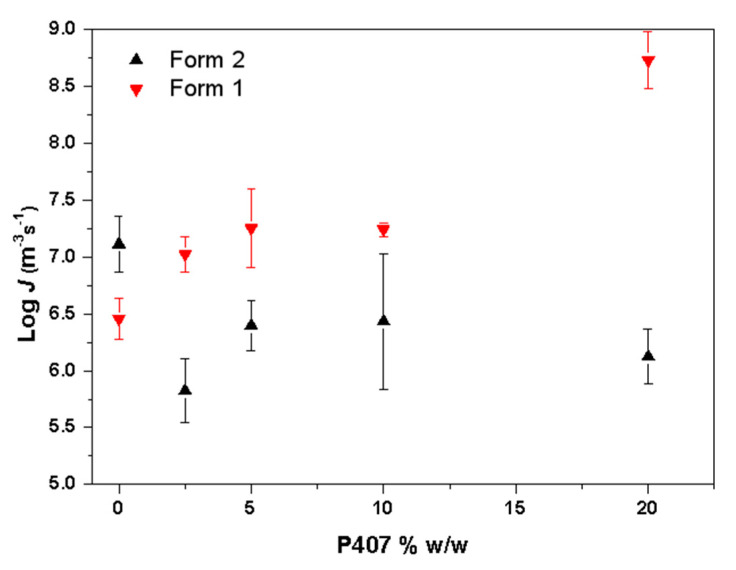
Crystal nucleation kinetics at 50 °C for CMZ Form 1 and Form 2 as a function of the P407 concentration.

**Figure 10 pharmaceutics-15-02164-f010:**
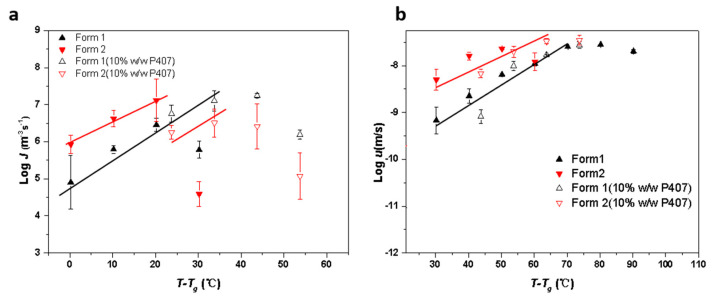
Nucleation rates (**a**) and growth rates (**b**) for CMZ with 10% *w*/*w* P407 added as a function of *T*–*Tg* (the lines are only a guide to the eye).

**Figure 11 pharmaceutics-15-02164-f011:**
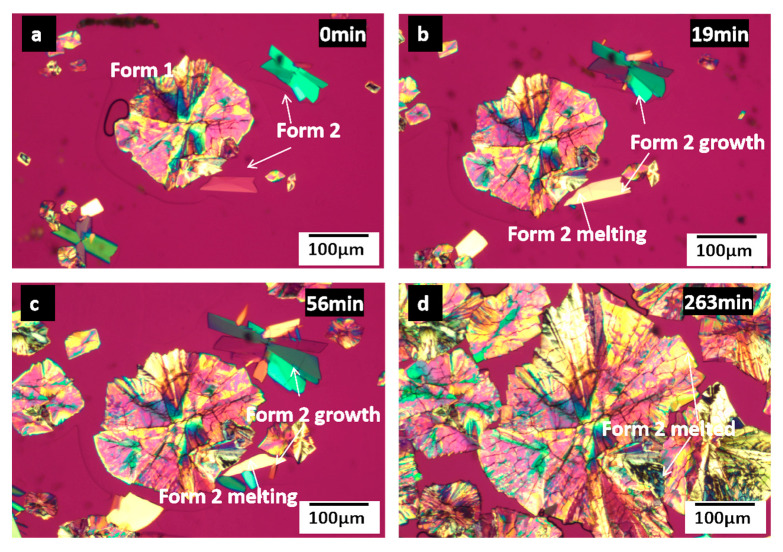
Competitive growth at 70 °C of CMZ polymorphs containing 10% *w*/*w* P407 at different times, 0 min (**a**); 19 min (**b**); 56 min (**c**); 263 min (**d**).

**Figure 12 pharmaceutics-15-02164-f012:**
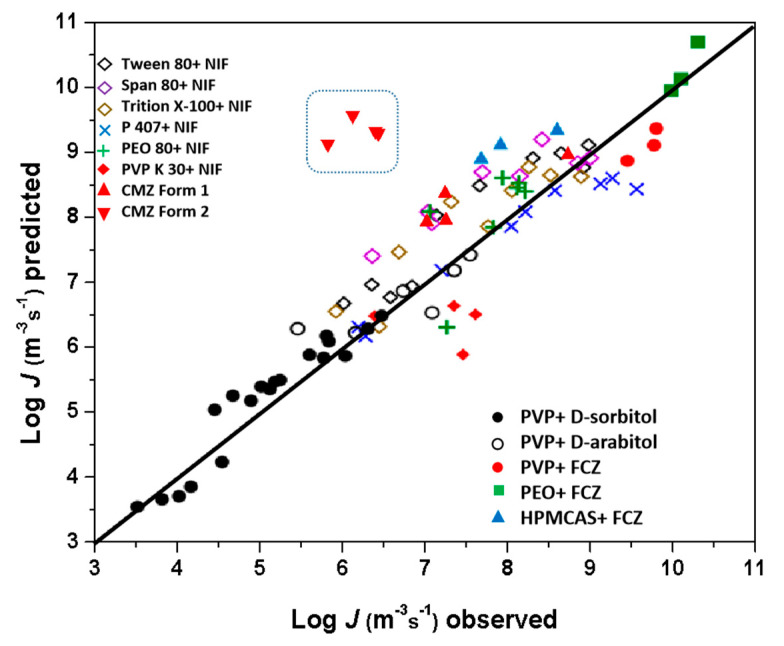
Observed vs. predicted nucleation rates for different systems: the plot was Reproduced with permission from [21], American Chemical Society, 2022.; the data for FCZ doped with different polymers are from [33]; the data for NIF doped with different surfactants are from [21]; D-sorbitol and D-arabitol doped with PVP are from [34]; the data for CMZ Form 1 and Form 2 doped with P407 are from this study.

## Data Availability

The data that support the findings of this study are available from the corresponding authors upon reasonable request.

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
