# Peer review of "Impact of Poloxamer on Crystal Nucleation and Growth of Amorphous Clotrimazole"

_pharmaceutics, 2023, doi:10.3390/pharmaceutics15082164_

Round 1

Reviewer 1 Report

In this work, the authors investigated the effect of the common surfactant poloxamer 407 on the crystallization of amorphous clotrimazole. The poloxamer showed different effects on nucleation of the clotrimazole polymorphs. The authors believe that their study provides insight into the effect of surfactants on the kinetics of drug nucleation and growth during melt crystallization. I think this work may be of interest to readers of Pharmaceutics. The manuscript can be recommended for publication after clarification of two important and several minor issues.

Important issues. 

1. What is the physical meaning of the nonmonotonic dependence of the crystal growth kinetics of Forms 1 and 2 on the P407 concentration at 80°C (Figure 8)? 

2. The conclusions are too general. Authors should list the specific results that were obtained in the work.

Minor issues. 

1. Keywords should not repeat the title of the article.

2. On pages 6, 7, 8 and 9, the square brackets used for references are missing.

3. Designations of drawings Figure 1a and Figure A1 make it difficult to read the paper. 

4. Avoid using abbreviations in conclusions. 

Author Response

Reviewer 1

Comments and Suggestions for Authors

In this work, the authors investigated the effect of the common surfactant poloxamer 407 on the crystallization of amorphous clotrimazole. The poloxamer showed different effects on nucleation of the clotrimazole polymorphs. The authors believe that their study provides insight into the effect of surfactants on the kinetics of drug nucleation and growth during melt crystallization. I think this work may be of interest to readers of Pharmaceutics. The manuscript can be recommended for publication after clarification of two important and several minor issues.

Important issues. 

  1. What is the physical meaning of the nonmonotonic dependence of the crystal growth kinetics of Forms 1 and 2 on the P407 concentration at 80°C (Figure 8)? 

Response: We thank the reviewer for the comment.In general, the crystal growth rates will increase and reach a plateau when the P407 loading increased,just like 50 ℃.It has been studied in our previous study(Mol. Pharmaceutics 2019, 16, 1385−1396; Mol. Pharmaceutics 2020, 17, 2064−2071).However, the crystal growth rates decreased after the P407 loading further increase, which means that other factor influences the growth rates. We suspect that the thermodynamic factor had a greater impact. The growth rates were determined by thermodynamics at temperatures closer to Tm, which was suppressed by the P407 content, shown in manuscript.

  1. The conclusions are too general. Authors should list the specific results that were obtained in the work.

Response: We thank the reviewer for the comment. Following the reviewer’s suggestion, we have changed it.

Minor issues. 

  1. Keywords should not repeat the title of the article.

Response: We thank the reviewer for the comment. We have made the correction.

  1. On pages 6, 7, 8 and 9, the square brackets used for references are missing.

Response: We thank the reviewer for the comments. We added square brackets used for references.

  1. Designations of drawings Figure 1a and Figure A1 make it difficult to read the paper. 

Response: We thank the reviewer for the comment. We have changed Figure A1 and Figure A2 to Figure S1 and Figure S1.

  1. Avoid using abbreviations in conclusions. 

Response: We thank the reviewer for the comment. We have made the correction.

Reviewer 2 Report

In this paper effect of surfactants on the melt crystallization of drugs was studied.

The authors studied the effect of the Poloxamer additive on the crystallization kinetics of amorphous Clotrimazole. Interestingly, they managed to detect different effects of the polymer on different polymorphs. The results obtained will be of interest to specialists in the field of crystal engineering, pharmaceuticals, and so on.

In my view, the manuscript can be published in the journal Pharmaceutics after corrections.

 Below are some suggestions and comments.

1. It is necessary to correct errors in the text, starting with the first sentence in the Introduction. References should be in square brackets.

2. Please remove the conclusions from the Introduction and move it to the Abstract or Conclusion.

3. The authors should expand the introduction and describe the effect of Poloxamer on the crystallization of other amorphous drug substances since such studies were carried out in large numbers.

4. It is necessary to justify the choice of Clotrimazole. Why is this drug?

5. In the Figures with DSC data, the indication of the direction of exo or endo effects should be shown.

6. The work lacks information on the particle size of the dispersed phase and the distribution of the polymer and drug in the solid phase. This affects the reproducibility of the results. The authors should comment on this issue. Alternatively, calculate the grain size from the PXRD data.

6. More discussion is needed for Figure 5. Data for pure drug corresponds to a mixture of polymorphs 1 and 2. However, the X-ray diffraction patterns of Clotrimazole with P407 lack the peaks characteristic of Form 2. Although Form 2 is present in Figure 6 even at high levels of P407. Need to comment.

7. According to DSC data, the polymer in the mixture has a melting point. So, are the polymer crystals visible in PLM?

Author Response

Reviewer 2

Comments and Suggestions for Authors

In this paper effect of surfactants on the melt crystallization of drugs was studied.

The authors studied the effect of the Poloxamer additive on the crystallization kinetics of amorphous Clotrimazole. Interestingly, they managed to detect different effects of the polymer on different polymorphs. The results obtained will be of interest to specialists in the field of crystal engineering, pharmaceuticals, and so on.

In my view, the manuscript can be published in the journal Pharmaceutics after corrections.

 Below are some suggestions and comments.

  1. It is necessary to correct errors in the text, starting with the first sentence in the Introduction. References should be in square brackets.

Response: We thank the reviewer for the comment. We have made the correction.

  1. Please remove the conclusions from the Introduction and move it to the Abstract or Conclusion.

Response: We thank the reviewer for the comment. Following the reviewer’s suggestion, we have changed it.

  1. The authors should expand the introduction and describe the effect of Poloxamer on the crystallization of other amorphous drug substances since such studies were carried out in large numbers.

Response: We thank the reviewer for the comment. Following the reviewer’s suggestion, we have added it.

  1. It is necessary to justify the choice of Clotrimazole. Why is this drug?

Response: We thank the editor for the comments. We added it.

  1. In the Figures with DSC data, the indication of the direction of exo or endo effects should be shown.

Response: We thank the editor for the comments. We added it.

  1. The work lacks information on the particle size of the dispersed phase and the distribution of the polymer and drug in the solid phase. This affects the reproducibility of the results. The authors should comment on this issue. Alternatively, calculate the grain size from the PXRD data.

Response: We thank the reviewer for the comment. The size of the drug crystals was measured by PLM and then calculated the growth rate of drug. Meanwhile the polymer did not crystallize during the drug crystallization at the evaluated temperature. Indeed, PXRD data can calculate the grain size; however, this study pays more attention on crystallizaiton of drug without polymer crystallization. When the polymer increaed to high content, the polymer will crystallize, this is in the scope of effect of drug on the crystallization of polymer. Further studies are still ongoing.

  1. More discussion is needed for Figure 5. Data for pure drug corresponds to a mixture of polymorphs 1 and 2. However, the X-ray diffraction patterns of Clotrimazole with P407 lack the peaks characteristic of Form 2. Although Form 2 is present in Figure 6 even at high levels of P407. Need to comment.

Response: We thank the reviewer for the comment. In Figure 5,Form 2 has a characteristic peak at 11.7â—‹,we can found that this peak exists in the CMZ doped with 2.5%-10% w/w P407.While this peak is not obvious in the CMZ doped with 20% w/w P407, which maybe attribute to the low content of Form 2 in the sample.It can be explained by the results from Figure 6. We added more discussion for this section.

  1. According to DSC data, the polymer in the mixture has a melting point. So, are the polymer crystals visible in PLM?

Response: We thank the reviewer for the comment. The concentration of the polymer is relatively low in this study, thus the polymer could not crystallize. When the polymer increaed to high content, the polymer will crystallize, this is in the scope of effect of drug on the crystallization of polymer. Further studies are still ongoing.

Reviewer 3 Report

The research topic is interesting and falls definitely within the scope of the journal.

However, the manuscript is plagued by several problems in its presentation. The main problem is that the authors provide very limited information on the interpretation of the observed trends. For example, it is impossible to understand what mean qualitatively the different mechanisms depicted in Fig. 13 as the accompanying text offers no such clue.  Authors suggest thermodynamic and kinetic factors affecting the crystallization process, but they provide very limited analysis of what is observed in the experiments.

) in Fig. 3 the legend describes “Tg values of amorphous…” while in Figure 3 has two panels and in Fig. 3a Heat flow vs. Temperature is presented.

Polymorphs Form1 and Form2 should be better explained in the introduction to make the manuscript more accessible to a general readership.

It is not clear if the data for the CMZ-P407 mixtures correspond to just one sample or several of them. Authors should make this clear in the methods section.

) Error bars should be added whenever possible, for example in Fig. 2. Figure legends should be clearer. For example, in the data sets of Fig. 2b it is not immediately clear which data correspond to CMZ and P407.

) Quality of text in axes in several figures is low (for example in Fig. 10). Variables are used for example J and u but these are never properly defined in the main text.

) It would be interesting if the authors could include a panel corresponding to time between 19 and 263 mins in Fig. 11.

) It is not clear what is meant by the file numbers used in the legend of Fig. 5.

The use of phrasing implying speculation should be limited. For example, “we suspect” in lines 21-22.

) What is meant practically by “determination temperature” (line 105)?

) Symbols for physical parameters should be presented in better format to be easier identified in main text (for example Tm in line 110; Tg in lines 118, 120 etc).

The manuscript requires a careful edit in English grammar and syntax. In some instances, the meaning is nuclear and the reading flow is not optimal.

Line 29: “The In” -> “In”

Line 34: “has been approved” -> “has approved”.

Line 54: “elevated” -> “evaluated”.

In main text notation CMZ should be define once first introduced (line 62).

Line 87: “were determined” -> “was determined”.

Line 133: “26, 27” -> “[26,27] (similar in lines 176, 180, 195, 197 etc). There seems to be a systematic error in the format of references after a point in the main text.

Line 135: “single cyrstals” -> “single crystal”.

The manuscript requites extensive editing with respect to its format and presentation and in some instances on the English language. 

Author Response

Reviewer 3

Comments and Suggestions for Authors

The research topic is interesting and falls definitely within the scope of the journal.

However, the manuscript is plagued by several problems in its presentation. The main problem is that the authors provide very limited information on the interpretation of the observed trends. For example, it is impossible to understand what mean qualitatively the different mechanisms depicted in Fig. 13 as the accompanying text offers no such clue.  Authors suggest thermodynamic and kinetic factors affecting the crystallization process, but they provide very limited analysis of what is observed in the experiments.

Response: We thank the reviewer for the comment. We deleted Figure 13. We added more discussion for results.

) in Fig. 3 the legend describes “Tg values of amorphous…” while in Figure 3 has two panels and in Fig. 3a Heat flow vs. Temperature is presented.

Response: We thank the reviewer for the comment. We changed it.

Polymorphs Form1 and Form2 should be better explained in the introduction to make the manuscript more accessible to a general readership.

Response: We thank the reviewer for the comment. Following the reviewer’s suggestion, we added more discussion for Form1 and Form2.

It is not clear if the data for the CMZ-P407 mixtures correspond to just one sample or several of them. Authors should make this clear in the methods section.

Response: We thank the reviewer for the comment. We changed it.

) Error bars should be added whenever possible, for example in Fig. 2. Figure legends should be clearer. For example, in the data sets of Fig. 2b it is not immediately clear which data correspond to CMZ and P407.

Response: We thank the reviewer for the comment. Only one experiment was conducted for different CMZ-P407 mixtures. Error bars have be added in other figures whenever possible. The legends were added in the Fig. 2b.

) Quality of text in axes in several figures is low (for example in Fig. 10). Variables are used for example J and u but these are never properly defined in the main text.

Response: We thank the reviewer for the comment. We changed it.

) It would be interesting if the authors could include a panel corresponding to time between 19 and 263 mins in Fig. 11.

Response: We thank the reviewer for the comment. We added it.

) It is not clear what is meant by the file numbers used in the legend of Fig. 5.

Response: We thank the reviewer for the comment. File numbers refer CCDC number of Form 1 and Form 2 in the Cambridge Crystallographic Data Centre (CCDC).

The use of phrasing implying speculation should be limited. For example, “we suspect” in lines 21-22.

Response: We thank the reviewer for the comment. We changed it.

) What is meant practically by “determination temperature” (line 105)?

Response: We thank the reviewer for the comment. The determination temperature is the desired temperature for measuring the crystal growth rate. We changed it.

) Symbols for physical parameters should be presented in better format to be easier identified in main text (for example Tm in line 110; Tg in lines 118, 120 etc).

Response: We thank the reviewer for the comment. We have changed it to italic style and made it to be easier identified.

The manuscript requires a careful edit in English grammar and syntax. In some instances, the meaning is nuclear and the reading flow is not optimal.

Line 29: “The In” -> “In”

Response: We thank the reviewer for the comment. We have made the correction.

Line 34: “has been approved” -> “has approved”.

Response: We thank the reviewer for the comment. We have made the correction.

Line 54: “elevated” -> “evaluated”.

Response: We thank the reviewer for the comment. We have made the correction.

In main text notation CMZ should be define once first introduced (line 62).

Response: We thank the reviewer for the comment. We have made the correction.

Line 87: “were determined” -> “was determined”.

Response: We thank the reviewer for the comment. We have made the correction.

Line 133: “26, 27” -> “[26,27] (similar in lines 176, 180, 195, 197 etc). There seems to be a systematic error in the format of references after a point in the main text.

Response: We thank the reviewer for the comments. We added square brackets used for references.

Line 135: “single cyrstals” -> “single crystal”.

Response: We thank the reviewer for the comment. We have made the correction.

Comments on the Quality of English Language

The manuscript requites extensive editing with respect to its format and presentation and in some instances on the English language.

Response: We thank the reviewer for the comment.The manuscript has been carefully edited by professional English editors. The corrections in the main text are highlighted in red

Reviewer 4 Report

The manuscript “Impact of Poloxamer on Crystal Nucleation and Growth of Amorphous Clotrimazole” describes the effect of a polymer surfactant Poloxamer P470 on the nucleation and growth of the drug Clotrimazole (CMZ) from amorphous solid dispersions.

It was found that the polymer increases the growth rate of two polymorphs of CMZ, while having a selective effect on the nucleation rate. I.e. the nucleation rate of Form 1 is increased by the polymer, while the nucleation of Form 2 is slightly depressed or unchanged. The study is interesting and is worth publishing after some questions and issues are addressed.

Please pay attention to Section2:

Please report how ASDs were prepared, at present only the preparation of the physical mixtures is described.

What does “crystallization was initiated at room temperature” mean? Does CMZ crystallize at room temperature?

Please describe how you estimate volumetric nucleation rate from the data obtained from the glass slides. How the sample volume was determined?

88: unclear sentence. Please describe the procedure properly.

92: better use nitrogen rather than N2. What was the flow rate?

92: “sample was loaded into a crimped pan”. Please rephrase, this sentence reads like you add something into an already crimped pan.

93: above melting temperature of what – poloxamer or clotrimazole? Please specify the exact temperature and time of the high temperature treatment.

98: Please specify the quenching conditions.

In the Results/Discussion

111: Please add a note that P407 and Pluronic F127 are essentially the same polymer.

Figure 2. Please express the X-axis in Figure 2b as polymer weight fraction, so it corresponds with Figure 2a captions.

Does the dependence of Tg on the polymer content follow the Gordon-Taylor equation?

Figure 7. Please add the lines connecting the data points.

Figure 8. Please plot both graphs at the same scale so the comparison is possible.

The authors explain the non-monotonous nature of the dependence of the growth rate of CNZ on polymer content by the effect of polymer on the melting points of the crystals, which in turns affects the thermodynamic driving force of crystallization. As such, please report the melting points of the crystals grown from solid dispersions.

Figure 9: Why data for the nucleation rate of Form 2 are approximated by the non-linear curve, whereas for Form 1 a linear curve is used? IMO the data for Form 2 can be approximated linearly as well.

The deviation of the nucleation rate of Form 2 CMZ from the straight line in Figure 12 can be interpreted by two options, either the observed nucleation rate of Form 2 is lower than predicted from the growth rate, or the growth rate is higher than what can be expected from the observed nucleation rates. What option is favored by the authors and why?

Is it possible that the apparent decrease in the nucleation rate of Form 2 is because some of Form 2 crystals convert to Form 1 in the early stages of the growth? Perhaps the authors may try using a higher magnification and check the evolution of the populations of Form 1 and Form 2 crystals in the early stages of the growth?

Please describe more thoroughly what is illustrated in Figure 13. IMO it does not do a good job of clarifying different effects of polymer on nucleation and growth. What is the green blob on the rightmost scheme?

Some minor language/editing issues

16: “a accelerating”

29: “The In”

46: “holt” “Tg”

54: “elevated”

136, 176, 177, 180, 195, 212, 213 etc… : Brackets around the citations are missing.

181: phrase “in the present study” is redundant

182-184: “more polymer was present in the stable form” – what does it mean? Can polymer penetrate into CMZ crystals? Or you mean “more polymer was present in the growth front of the stable form”?

222: poymer

Author Response

Reviewer 4

Comments and Suggestions for Authors

The manuscript “Impact of Poloxamer on Crystal Nucleation and Growth of Amorphous Clotrimazole” describes the effect of a polymer surfactant Poloxamer P470 on the nucleation and growth of the drug Clotrimazole (CMZ) from amorphous solid dispersions.

It was found that the polymer increases the growth rate of two polymorphs of CMZ, while having a selective effect on the nucleation rate. I.e. the nucleation rate of Form 1 is increased by the polymer, while the nucleation of Form 2 is slightly depressed or unchanged. The study is interesting and is worth publishing after some questions and issues are addressed.

Please pay attention to Section2:

Please report how ASDs were prepared, at present only the preparation of the physical mixtures is described.

Response:We thank the reviewer for the comment. Following the reviewer’s suggestion, we added it.

What does “crystallization was initiated at room temperature” mean? Does CMZ crystallize at room temperature?

Response:We thank the reviewer for the comment. “crystallization was initiated at room temperature” mean “The amorphous samples were hold for several minutes or hours at room temperature for nucleation, and then transferred to high temperature for growth rates determination.”We changed it.

Please describe how you estimate volumetric nucleation rate from the data obtained from the glass slides. How the sample volume was determined?

Response:We thank the reviewer for the comment. Following the reviewer’s suggestion, we added it.

88: unclear sentence. Please describe the procedure properly.

Response:We thank the reviewer for the comment. Following the reviewer’s suggestion, we changed it.

92: better use nitrogen rather than N2. What was the flow rate?

Response:We thank the reviewer for the comment. Following the reviewer’s suggestion, we changed it and added the flow rate.

92: “sample was loaded into a crimped pan”. Please rephrase, this sentence reads like you add something into an already crimped pan.

Response:We thank the reviewer for the comment. Following the reviewer’s suggestion, we changed it.

93: above melting temperature of what – poloxamer or clotrimazole? Please specify the exact temperature and time of the high temperature treatment.

Response:We thank the reviewer for the comment. Following the reviewer’s suggestion, we added it.

98: Please specify the quenching conditions.

Response:We thank the reviewer for the comment. Following the reviewer’s suggestion, we added it.

In the Results/Discussion

111: Please add a note that P407 and Pluronic F127 are essentially the same polymer.

Response:We thank the reviewer for the comment. Following the reviewer’s suggestion, we added it.

Figure 2. Please express the X-axis in Figure 2b as polymer weight fraction, so it corresponds with Figure 2a captions.

Response:We thank the reviewer for the comment. Following the reviewer’s suggestion, we changed it.

Does the dependence of Tg on the polymer content follow the Gordon-Taylor equation?

Response:We thank the reviewer for the comment. Following the reviewer’s suggestion, we calculated theoretical Tg according to Gordon-Taylor equation.

where Tg mix is the theoretical glass transition temperature of the drug–polymer blend and wd and wp and Tgd and Tgp are the weight fractions and glass transition temperatures (in Kelvin) of the pure drug and polymer, respectively. K is a constant, which can be further expressed as:

The results are shown in following table.The calculated theoretical Tg is higher than the observed value.

P407 content

Observed(K)

Gordon-Taylor equation(K)

2.5

299.47

298.34

5

290.36

294.06

10

279.26

286.01

20

265.94

271.64

Figure 7. Please add the lines connecting the data points.

Response:We thank the reviewer for the comment. Following the reviewer’s suggestion, we added it.

Figure 8. Please plot both graphs at the same scale so the comparison is possible.

Response:We thank the reviewer for the comment. Following the reviewer’s suggestion, we changed it.

The authors explain the non-monotonous nature of the dependence of the growth rate of CNZ on polymer content by the effect of polymer on the melting points of the crystals, which in turns affects the thermodynamic driving force of crystallization. As such, please report the melting points of the crystals grown from solid dispersions.

Response:We thank the reviewer for the comment. Following the reviewer’s suggestion, we added it.

Figure 9: Why data for the nucleation rate of Form 2 are approximated by the non-linear curve, whereas for Form 1 a linear curve is used? IMO the data for Form 2 can be approximated linearly as well.

Response:We thank the reviewer for the comment. Following the reviewer’s suggestion, we changed it.

The deviation of the nucleation rate of Form 2 CMZ from the straight line in Figure 12 can be interpreted by two options, either the observed nucleation rate of Form 2 is lower than predicted from the growth rate, or the growth rate is higher than what can be expected from the observed nucleation rates. What option is favored by the authors and why?

Response:We thank the reviewer for the comment. In my opion, the observed nucleation rate of Form 2 is lower than predicted from the growth rate. The nucleation rate of Form 2 decreased with increasing the P407, while the growth rates increased with increasing the P407. The nucleation of Form 2 is influenced by P 407 and Form 1.

Is it possible that the apparent decrease in the nucleation rate of Form 2 is because some of Form 2 crystals convert to Form 1 in the early stages of the growth? Perhaps the authors may try using a higher magnification and check the evolution of the populations of Form 1 and Form 2 crystals in the early stages of the growth?

Response:We thank the reviewer for the comment. This will be a great addition to make our

discussion more thorough. We agree with the reviewer that the apparent decrease in the nucleation rate of Form 2 is because some of Form 2 crystals convert to Form 1 in the early stages of the growth. We tried using a higher magnification and check the evolution of the populations of Form 1 and Form 2 crystals in the early stages of the growth.However,it is hard to observed the evolution in the early stages.We only observed the transformation of Form 2 to Form 1 when the crystals grow to a large size, and we compared the transformation rate for sample with and without P407,we added it to the supportting information.The addtion of P407 could induce a faster transformation the Form 2 crystals convert to Form 1 in the very early stages.

Please describe more thoroughly what is illustrated in Figure 13. IMO it does not do a good job of clarifying different effects of polymer on nucleation and growth. What is the green blob on the rightmost scheme?

Response: We thank the reviewer for the comment. We deleted Figure 13.

Comments on the Quality of English Language

Some minor language/editing issues

16: “a accelerating”

Response: We thank the reviewer for the comment. We have made the correction.

29: “The In”

Response: We thank the reviewer for the comment. We have made the correction.

46: “holt” “Tg”

Response: We thank the reviewer for the comment. We have made the correction.

54: “elevated”

Response: We thank the reviewer for the comment. We have made the correction.

136, 176, 177, 180, 195, 212, 213 etc… : Brackets around the citations are missing.

Response: We thank the reviewer for the comments. We added square brackets used for references.

181: phrase “in the present study” is redundant

Response: We thank the reviewer for the comment. We have made the correction

182-184: “more polymer was present in the stable form” – what does it mean? Can polymer penetrate into CMZ crystals? Or you mean “more polymer was present in the growth front of the stable form”?

Response: We thank the reviewer for the comment. It means “more polymer was present in the growth front of the stable form”.We have made the correction.

222: poymer

Response: We thank the reviewer for the comment. We have made the correction.

Round 2

Reviewer 2 Report

After reading the manuscript, I concluded that the authors corrected most of the reviewers comments.

Author Response

Response: We thank the reviewer for the comment. 

Reviewer 3 Report

Authors have partially addressed some of the comments raised with respect to the original version and the manuscript has been improved. Their response letter is quite blur with sentences like “it has been fixed” while no further information is given nor is clear if indeed there has been a change in the text. The manuscript still requires revisions before being in publishable form.

) In Fig. 7 data points are connected through lines. Do the lines correspond to some expression and have physical interpretation? Are they just guide for the eye? Authors should clarify this. Same for Fig. 8, how the solid curves lines are obtained? What is their purpose? Also, the x-axis of Fig. 8 should start from zero. Then in Fig. 9 are the lines the result of best linear fits? Again, do the fitting parameters have a physical correspondence? Authors could report the fitting results in tables as Appendix. Again in Fig. 10, how the lines are selected and what do they represent? Finally, in Fig. S3 the red and black lines cannot be best linear fits as they over- and under-represent, respectively, the behavior of the points.

) I have written that authors should avoid using expressions like “suspect” especially in the abstract but the wording seems unchanged (for example line 20 in abstract). In Fig. 2 I still cannot understand the correspondence of the blue squares. Why are they different from the black and red ones? In some instances (for example the third blue point) seems to be higher by at least 10oC compared to the analogous red.

) In lines 72-76 the syntax is confusing. It is not clear if the authors refer to results obtained in the present manuscript or to past works. If it is the latter case, then the corresponding references should be provided.

The manuscript is still plagued by syntax errors:

Line 72: “form” -> “from”; “glass transition changing” -> “change in the glass transition”.

Line 94: “were determined” -> “was determined”.

Line 105: “sample was 2-5 mg” -> “2-5 mg sample was”.

Line 124: “increase” -> “increases”.

Line 128: “can be transform” -> “can be transformed”.

Line 167: “may be attributed”.

Line 298: “form 1” -> “Form 1”.

The syntax has been improved, however the manuscript still requires a careful edit in English. 

Author Response

Comments and Suggestions for Authors

Authors have partially addressed some of the comments raised with respect to the original version and the manuscript has been improved. Their response letter is quite blur with sentences like “it has been fixed” while no further information is given nor is clear if indeed there has been a change in the text. The manuscript still requires revisions before being in publishable form.

) In Fig. 7 data points are connected through lines. Do the lines correspond to some expression and have physical interpretation? Are they just guide for the eye? Authors should clarify this. Same for Fig. 8, how the solid curves lines are obtained? What is their purpose? Also, the x-axis of Fig. 8 should start from zero. Then in Fig. 9 are the lines the result of best linear fits? Again, do the fitting parameters have a physical correspondence? Authors could report the fitting results in tables as Appendix. Again in Fig. 10, how the lines are selected and what do they represent? Finally, in Fig. S3 the red and black lines cannot be best linear fits as they over- and under-represent, respectively, the behavior of the points.

Response: We thank the reviewer for the comment. We added the lines connecting the data points, they are just guide for the eye. We think that the addition of lines is easier to compare the results. The x-axis of Fig. 8 is changed and start from zero.

) I have written that authors should avoid using expressions like “suspect” especially in the abstract but the wording seems unchanged (for example line 20 in abstract). In Fig. 2 I still cannot understand the correspondence of the blue squares. Why are they different from the black and red ones? In some instances (for example the third blue point) seems to be higher by at least 10oC compared to the analogous red.

Response: We thank the reviewer for the comment. We fixed expressions like “suspect” in the manuscript. In Figure 2, We added “(black squares and red squares indicate the melting points of the drug and P407 are from ref. [26]; the data presented with blue squares (drug) and green squares (P407)were obtained in this work)” and changed the Figure 2a.

The results from this study is similar to the reference,however, In some instances (for example the third blue point),our result(50.6℃) is higher than the reference(41.9℃.For this point, the P407 content is 10% w/w ,the heat fusion is small,and we think this may influence the determination of the melting point of P407, especially for different group.

) In lines 72-76 the syntax is confusing. It is not clear if the authors refer to results obtained in the present manuscript or to past works. If it is the latter case, then the corresponding references should be provided.

Response: We thank the reviewer for the comment. We refer to results obtained in the present manuscript. We clarify this in the manuscript.

The manuscript is still plagued by syntax errors:

Line 72: “form” -> “from”; “glass transition changing” -> “change in the glass transition”.

Response: We thank the reviewer for the comment. We have made the correction.

Line 94: “were determined” -> “was determined”.

Response: We thank the reviewer for the comment. We have made the correction.

Line 105: “sample was 2-5 mg” -> “2-5 mg sample was”.

Response: We thank the reviewer for the comment. We have made the correction.

Line 124: “increase” -> “increases”.

Response: We thank the reviewer for the comment. We have made the correction.

Line 128: “can be transform” -> “can be transformed”.

Response: We thank the reviewer for the comment. We have made the correction.

Line 167: “may be attributed”.

Response: We thank the reviewer for the comment. We have made the correction.

Line 298: “form 1” -> “Form 1”.

Response: We thank the reviewer for the comment. We have made the correction.

Comments on the Quality of English Language

The syntax has been improved, however the manuscript still requires a careful edit in English.

Response: We thank the reviewer for the comment.The manuscript has been carefully edited by professional English editors. The corrections in the main text are highlighted in red

Round 3

Reviewer 3 Report

The manuscript has been improved and it now very close to being in publishable form.

The last major comment is that it should be clarified in all figure legends if the lines/curves shown are just guide for the eye or a result of some kind of fitting function. For example, I have trouble believing that the black curves in Fig. 8 are just guide for the eye. Same for Fig. 9 where both the red and black lines seem like best linear fits to the scattered data.

If the lines are just guide for the eye, I would suggest removing them from Fig. S3 as they appear misleading and do not follow the data trends.

Line 16: “combination effect” -> “combined effect”

Line 70: “had been” -> “has been”, “through change” -> “through the change”

Line 88: “Crystallne” -> “Crystalline”

The manuscript has been improved in this respect but in case of publication a final polishing is required. 

Author Response

Comments and Suggestions for Authors

The manuscript has been improved and it now very close to being in publishable form.

The last major comment is that it should be clarified in all figure legends if the lines/curves shown are just guide for the eye or a result of some kind of fitting function. For example, I have trouble believing that the black curves in Fig. 8 are just guide for the eye. Same for Fig. 9 where both the red and black lines seem like best linear fits to the scattered data.

If the lines are just guide for the eye, I would suggest removing them from Fig. S3 as they appear misleading and do not follow the data trends.

Response: We thank the reviewer for the comment. In figures 7 and 10, we added clarification that the lines/curves shown are just guide for the eye.We removed them from figures 8, 9 and S3 as they may appear misleading.

Line 16: “combination effect” -> “combined effect”

Response: We thank the reviewer for the comment. We have made the correction.

Line 70: “had been” -> “has been”, “through change” -> “through the change”

Response: We thank the reviewer for the comment. We have made the correction.

Line 88: “Crystallne” -> “Crystalline”

Response: We thank the reviewer for the comment. We have made the correction.

Comments on the Quality of English Language

The manuscript has been improved in this respect but in case of publication a final polishing is required.

Response: We thank the reviewer for the comment.The manuscript has been carefully edited by professional English editors. The corrections in the main text are highlighted in red